# Current Denture Loss in Geriatric Facilities

**DOI:** 10.3390/medicines9110053

**Published:** 2022-10-26

**Authors:** Miki Endo, Nami Nakayama, Miki Yamada, Yosuke Iijima, Shunsuke Hino, Kiyoko Ariya, Norio Horie, Takahiro Kaneko

**Affiliations:** 1Department of Oral and Maxillofacial Surgery, Saitama Medical Center, Saitama Medical University, Saitama 350-8550, Japan; 2Attaka no ie, Special Elderly Care Home, Social Welfare Corporation, Shiki Welfare Association, Saitama 350-0003, Japan

**Keywords:** denture loss, geriatric facility, older adults

## Abstract

Purpose: Denture loss is still being reported as a problem in geriatric facilities, although losses seem less frequent than in the last decade. However, there have been no reports that have examined recent losses of dentures in detail. The aim of this study was to clarify the actual situation of recent denture loss, together with the denture loss rate in Japan. Materials and methods: This retrospective study investigated the number of cases of denture loss, the denture loss rate for denture wearers, and the details of losses in geriatric facilities during the 1-year period from 1 April 2020 to 31 March 2021. Results: Eleven special elderly nursing homes and four group homes participated in this research. The number of residents from each was 315 and 40 and the number of denture wearers was 165 and 33, respectively (*p* < 0.001). The loss of dentures was found in one case from a special elderly nursing home and in one case from a group home. The loss rate for denture wearers was 1.01% in total, with 0.61% for special elderly nursing homes and 3.03% for group homes, with no significant differences between the two types of facilities. Conclusion: In geriatric facilities in Japan, the current 1-year denture loss rate for denture wearers was 1.01%. This seems to represent a considerable decrease when compared with the previous report. Further, proper denture management and staff efforts appear to have contributed to a reduction in denture loss against a background of promoting oral healthcare.

## 1. Introduction

Along with an increase in the elderly population, the number of individuals with missing teeth is increasing, and many older adults also use prosthetics. Mastication, pronunciation, and facial aesthetics can be restored by the prosthetic treatment of missing teeth. Among the prosthetic procedures available, removable dentures are in wide use as they can properly, simply, and reliably recover deficits in the dentition at low cost. As the elderly population increases, so too does the number of residents in care facilities, and a relatively higher proportion of denture wearers are, unsurprisingly, residents of care facilities and geriatric wards [1].

One of the troublesome problems with denture handling in care facilities and wards are the loss of dentures [2]. This represents a sporadic problem. Loss of dentures and the potential countermeasures have been a matter of discussion for some time. In 1986, Harrison noted that a university hospital experienced 79 denture losses in a 30 month period [3]. In 2007, Michaeli et al. reported 30 cases of denture loss in 8 months at facilities for older adults and inpatient wards, although the number of subjects was unknown [4]. Further, Doshi et al. state that denture loss is still a major problem these days [2].

Japan, along with the rest of the world, is facing an increasingly aging population, and the number of users in facilities for the elderly has, likewise, been increasing [5]. We are also engaged in oral health care at geriatric facilities, and also recently encountered a case of denture loss that made us interested in the issue of denture loss at geriatric facilities. Although previous reports have reported a relatively large number of denture losses, our feeling was that the number of denture losses is not as high as previously reported [4,6,7]. Additionally, no specific investigations into the rate of denture loss in geriatric facilities have recently been conducted. Loss of dentures is a problem related to the quality of the management system at the facility. Quantifying current denture loss is therefore meaningful. The aim of this study was to specifically determine both the number and rate of denture loss and to clarify the actual situation of denture loss in geriatric facilities in Japan.

## 2. Materials and Methods

This study was designed in accordance with the Declaration of Helsinki and also the Good Clinical Practice guidelines; further, it was approved by the Ethics Committee of Saitama Medical Center, Saitama Medical University (2021-184). To address the research purpose outlined above, the investigators designed and implemented this retrospective study.

The 1-year denture loss situation from 1 April 2020 to 31 March 2021 was retrospectively investigated for older adults (≥65 years old) in geriatric facilities (i.e., special elderly nursing homes and group homes). Special elderly nursing homes are facilities providing end-of-life care to the elderly [8], while group homes are facilities for people with dementia to live together in small groups—where the residents can handle communal living and are relatively capable of activities of daily living [9]. Residents of both types of the facilities are regularly examined by a medical doctor; further, oral health care is conducted by a dental hygienist once a week, and a dentist is available at least twice a month. All the facilities searched were those in which the staff at the authors’ clinic were involved in providing oral healthcare. The data used in this search were based on the oral health care records and a questionnaire survey obtained from each facility.

The searched items were the number of lost dentures and the denture loss rate. When calculating the denture loss rate, the population was restricted to denture wearers. A denture wearer was defined as a resident who owned and actually used the dentures. Those who did not own dentures or those who owned dentures but did not use them were excluded as per the exclusion criteria. Although the number of residents changes continually, the next resident in each facility would arrive within a few days, as such the occupancy rate of each facility was considered to be 100% throughout the year. The number of denture wearers would likewise change, but for the sake of convenience, the number of denture wearers as of 31 March 2021 was used in this search. In cases of lost dentures, the details were also investigated.

All statistical analyses were performed using js-STAR version 9 software (http://www.kisnet.or.jp/nappa/software/star/index.htm; accessed on 26 April 2022). Fisher’s exact test was used to compare the rate of denture wearers and the denture loss rate for denture wearers between the two types of facilities. The level of significance was set at 5% (*p* < 0.05).

## 3. Results

The study cohort of 355 elderly individuals included 315 residents of 11 special elderly nursing homes and 40 residents of 4 group homes. Each facility met the relevant facility standards, and no differences were apparent between facilities. Table 1 shows the number of residents, the rate of denture wearer, the number of dentures lost, and the denture loss rate for denture wearers at the two types of geriatric facility. Overall, 198 residents were denture users (198/355; 55.8%), this was broken down as follows: 33 denture wearers in group homes (33/40; 82.5%) and 165 denture wearers in special elderly nursing homes (165/315; 52.4%). A significantly higher proportion of residents in group homes were thus denture wearers. Loss of dentures was identified for one case in a special elderly nursing home and also one case in a group home. Denture loss rates for denture wearers were thus 0.61% and 3.03% for special elderly nursing homes and group homes, respectively, and 1.01% for the total cohort. No significant difference in denture loss rate was evident between denture wearers at the two types of facilities. As for the details of the two missing dentures, a resident in a special elderly nursing home had displayed highly disturbed and agitated behaviour, as well as displayed psychological symptoms of dementia, and was, as a result, unable to control her anger. This may have meant that she spit out her lower partial denture. A resident in a group home was also unable to manage her dentures well due to dementia. On the day she was admitted, she noticed that she was missing a maxillary full lower denture in her mouth. Table 2 shows the summary of the two cases of denture loss. While the possible causes differed, both cases involved individuals with dementia who were able to remove the dentures themselves.

## 4. Discussion

In this study, 52.4% and 82.5% of residents in the two types of geriatric facility (special elderly nursing homes and group homes) were denture wearers, with a significantly higher rate of denture wearers in group homes. This was most like due to the fact that many residents in special nursing homes need more advanced nursing care, and many residents thus could not wear dentures for physical or mental reasons.

Loss of dentures has always been a problem and is not new; additionally, some facilities have reported the number of lost dentures in their facilities. However, there are no reported loss rates. The loss of two dentures in one year, one from a special elderly nursing home and one from a group home, yielded loss rates of 0.61% for special elderly nursing homes and 3.03% for group homes—therefore, showing no significant difference. The overall denture loss rate for denture wearers in all 15 facilities, including 11 special elderly nursing homes and 4 group homes, participating in the present study was 1.01%. In this report, no more than one denture was lost within the same facility, and the loss of dentures within a facility was considered to be a rare incident. While the result obviously cannot reflect the current situation in other countries, the current situation of denture loss in general facilities in Japan may have been indicated in this result as the facilities participating in this research were standard facilities.

Mann and Doshi listed the five most common reasons for denture loss and proposed four methods in order to reduce losses [10,11]. These five common reasons for denture loss were: (1) The dentures were wrapped in tissue and left on meal trays; (2) the dentures were hidden in linen; (3) the dentures were mistaken for rubbish and thrown away; (4) the dentures were lost in transit between wards or theatre; and (5) the dentures were disposed of following an episode of vomiting when the dentures were expelled at the same time [11]. These five reasons cover cases that are extremely common in clinical practice.

The four methods to reduce denture loss proposed by Mann and Doshi were as follows: (1) staff training; (2) dentures pots; (3) denture labelling; and (4) raising awareness among patients and their families [11]. Regarding the first point, to prevent the loss of dentures, Michaeli et al. stated that the development of appropriate protocols and staff training is necessary in order to minimise these risks and to improve the standard of denture care [4]. In addition, external staff from multiple disciplines visit the rooms of facility residents. Mann and Doshi stated that training and education are also required for porters transporting patients and cleaning staff [11].

With regard to the second point, Mann and Doshi advocated placing dentures that were not in use into designated pots [11]. Deciding on a specific place to put dentures prevents the dentures from being inadvertently thrown away as garbage. Regarding the third point, Harrison emphasised writing names on dentures as a way of reducing denture loss [3,6]. Writing a name is also particularly useful when residents and patients go outside of the facility. Mann and Doshi noted that eight dentures found in the linen room of their Trust had been able to be returned due to name labelling [11]. The facilities included in the present research used different containers, but all facilities did store the dentures in individually named containers. The fourth point involves raising awareness among patients and family, advising residents to avoid wrapping dentures in tissues or linen to prevent dentures accidentally being discarded as garbage. It is suggested that unused dentures be taken home. If none of the rest of the family are denture wearers, the importance of dentures to edentulous patients can be difficult to fully appreciate.

The reason for the very low rate of denture loss in this study was attributed to the appropriate measures, as suggested by Mann and Doshi [11], that are increasingly being adopted at each facility. The tireless efforts of facility staff were also clearly important. However, the background to this is suggested to be recognition of the importance of oral health care and a growing momentum toward the promotion of oral health care. In recent years, the usefulness of oral health care has been emphasised in various situations, such as in the prevention of aspiration pneumonia and the treatment of oral mucositis, which is an adverse event resulting from cancer chemotherapy and radiotherapy [7,12]. Looking at the process of recognising the utility of adequate oral healthcare, such as searching for oral healthcare using search words in PubMed/MEDLINE, the number of articles has increased significantly since around 2010. In Japan, perioperative oral function management for cancer patients undergoing surgery or chemotherapy were first introduced into the national health insurance system in 2012, and interest in oral health care has since been increasing [13]. This growing interest in the oral environment is considered to be improving knowledge about oral health care and providing motivation for medical staff such as nurses and caregivers, thus greatly contributing to the prevention of denture loss [1,14].

Two cases of denture loss were found in this study. Both the residents involved had dementia and were able to remove their dentures independently. The cause of each loss was most likely that the resident removed the dentures themselves, then misplaced or forgot about the dentures. The progression of dementia seems likely to have a significant impact on denture management [15].

If dentures are known to have been lost in a facility, some issues must be considered. First, is the denture really missing from the vicinity of the resident? A careful search needs to be conducted, as dentures may often be hidden among the patient’s clothing, and the irregular shape of the dentures may then actually injure the patient in some cases. In rare cases, an emergency situation arises where a resident swallows the dentures, creating a potentially life-threatening situation. Aspirated dentures are often caught in the oesophagus and require urgent endoscopic or surgical removal under hospital admission [16]. Accidental ingestion of dentures may not cause immediate symptoms, so caution is required [17].

Loss of dentures places a heavy burden on both residents and patients. Some time is needed to create a new set of dentures, and aesthetics and oral functions such as pronunciation and mastication will be impaired during that time. Dentures that have been used by an individual for a long time are familiar, and Michaeli et al. reported the difficulty of replacing dentures that had been used for years with new dentures [4]. Mann and Doshi especially emphasised that the appearance of an individual without their normal dentures can cause mental distress to residents [11]. Lost dentures are generally remanufactured, but cognitive decline can make dental treatment difficult [18].

When a denture is lost to a resident or inpatient, a key problem is where the responsibility for the loss lies; that is, who shoulders the cost burden of remanufacturing. Financial complications of denture loss can be significant as the number of denture losses increases [11]. Of course, if the facility or hospital is responsible for the loss of the denture, the cost will be borne by the facility or hospital. Even if the resident or patient is responsible, the facility and hospital are responsible for proper management in many cases.

Some facilities do not allow dentures to be worn except when eating, for fear of loss. This idea certainly has some merit, but dentures contribute to the activation of synaptic/neuronal dysfunction in the brain [19]. In addition, nocturnal wearing of dentures has been shown to be effective for elderly patients with sleep apnoea [10]. Not using dentures except during meal times would thus represent a step-down method from a quality-of-life perspective.

This study showed some limitations. First, the sample sizes were small because the study was conducted at geriatric facilities related to a single institute. The larger the sample size, the more accurate the results are likely to be. However, we believe that this study successfully demonstrated the fact that denture loss is not as frequent as was thought before. Second, this study did not take into account the degree of care and the cognitive function of the subjects. The question of how the care level and cognitive function of residents are related to denture loss is an issue for future study. Third, this study surveyed a period of one year. Looking back, the link between the promotion of oral healthcare and the transition of denture loss could possibly be better shown in a longer study period.

What we are conveying in this study is only the current status of denture loss in Japan. It is true that there are countries in the world with various socioeconomic conditions. However, dentures are often the simplest and most effective prosthesis to improve oral function in any socioeconomic environment. We hope this study will be of some help in preventing denture loss.

In conclusion, the 1-year denture loss rate for denture wearers in geriatric facilities in current situation of Japan was 1.01%. This value seems to have decreased considerably compared with the previous report, and proper denture management and staff efforts appear to contribute to marked reductions in denture loss against the background of promoting oral healthcare.

## Figures and Tables

**Table 1 medicines-09-00053-t001:** Denture wearer rate and denture loss rate in the two types of geriatric facility.

	Number of Residents	Number of Denture Wearers (%)	*p* Value	Number of Lost Dentures	Denture Loss Rate for Denture Wearers (%)	*p* Value
Special elderly nursing homes (11 facilities)	315	165 (52.4)	<0.001	1	0.61	0.31
Group home (4 facilities)	40	33 (82.5)		1	3.03	
Total	355	198 (55.8)		2	1.01	

**Table 2 medicines-09-00053-t002:** Summary of the cases of denture loss.

Case in a Special Elderly Nursing Home	
Resident’s general status	Mainly remaining in bed during the day with dementia
Handling of dentures	Unable to put in, but able to remove by herself
Type of lost denture	Upper partial denture
Possible causes of lost dentures	Accidentally discarded during bed making or excrement disposal
Case in a group home	
Resident’s general status	Mainly remaining in bed during the day with dementia
Handling of dentures	Able to put in and remove dentures by herself
Type of lost denture	Lower full denture
Possible causes of lost dentures	Lost while organising luggage at the time of admission

## Data Availability

Not applicable.

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
