# Peer review of "Current Denture Loss in Geriatric Facilities"

_medicines, 2022, doi:10.3390/medicines9110053_

Round 1

Reviewer 1 Report

Please check the attached PDF with reviewer's comments.

Author Response

Thank you for your helpful review. The points that the reviewer raised have been corrected as follows.

Responses to the comments.

The results should be more statistically analyzed as it is not clearly give impact to the paper and show strength of the study (abstract section).

In accordance with the reviewer’s suggestion, Significant differences were added at the statistical analysis (Line 18 and 20).

This may be better written in the result. Usually in summary has to be conducted whether it is high or low and whether it is relevant or following the result from existing research (abstract section).

The reviewer makes a good point. However, the numbers of the percentage of dentures lost are very convincing in understanding the current situation. That is why we have included the % of denture loss in the conclusion section.

This method was a bit inappropriate due to the occupancy rate is not similar pre- and post-analysis. This is because the total number of resident at the beginning might be not the same even through it is considered as 100% which shows that the validity of the result is a bit off (Materials and Methods).

The reviewer's point is well taken. However, we believe that we have no choice in this report. This issue will be addressed in the next report, which is expected to have a large number of participants.

The other thing is, the 2 facilities has different type of geriatric care or home. This should be better if can list out the inclusion and exclusion criteria of the denture wearers that include in the study for both facilities (Materials and Methods).

In accordance with the reviewer’s suggestion, exclusion criteria were further clarified (Line 72).

A bit confuse on the explanation here regarding the total denture. Can rephrase the sentence or explain in clear sentence. As when refer to the respondents it is not good proportion. Maybe can increase the respondents to get better result that represent the population (Results).

In accordance with the reviewer’s suggestion, I have made the numbers in the text clearer (Line 91). Note that 32 in the text was an error for 33. This has been corrected.

Maybe this can be presented in percentage especially for the reason of missing denture to be more scientific presented for journal. To be able to present in percentage that is why the subjects should be increase to represent the population (Table 2).

As the reviewer pointed out, the population was small and there was a total of two cases of denture loss, so the percentages are not shown in this report.

This discussion is not suitable to present as this study only involved 2 facilities as compared with 15 facilities from previous study (Discussion section).

The text pointed out by the reviewer was difficult to understand, so we have rewritten it for clarity (Line 119).  

I’m not sure about the meaning of this sentence as by my understanding it seems not currently similar to results(Discussion section).

We will delete this statement. 

General comment:

  1. The result should be presented in well analyzed and more scientific presented.

  In accordance with the reviewers' comments, we have revised the manuscript to the best possible form in this revise.  

  1. In order to have better result, this study need to used same type of facilities unless this is the only facilities available at the study area. But this have to be informed in the methods.

We are aware that the population of group homes is small in this report. Please allow us increase the number of participants in the future.

  1. preferably the facilities should be increase or the number of denture wearers should be same pre and past data collection

As the reviewer pointed out, a large population is needed to improve the accuracy of the study. This will be an issue to be addressed in the future. 

  1. Discussion is well explained but the sentences should be more understandable as some of it slightly not similarly presented in results. Some information in discussion may be written in the result if it is the percentage of the result’s findings.

In accordance with the reviewer’s suggestion, I rewrote the discussion section text (Line 115).

We have revised this paper to the best of our ability. Thank you for your review and the helpful suggestions. We hope that the revised version of the manuscript will now be considered suitable for publication in medicines.

Sincerely,

Reviewer 2 Report

Article: Current denture loss in geriatric felicities.

The objective of this article is to update the data concerning the loss of dentures in retirement homes and specialized establishments welcoming mentally impaired elderly patients. It sheds light on the level of care for these elderly patients living in institutions in Japan.

However, the information concerning the patients included in the study is insufficient.

A questionnaire and the consultation of patient files do not allow to realize exactly the oral state of patients. Also in order to enhance this study, the collection of available medical data would be desirable, supplemented by a mini dental check-up. This as part of overall patient care. Thus, following this examination, the finding of the actual loss of the dental prosthesis remains necessary On the other hand the effective appreciation of the functional value and the aesthetics of the prosthesis of use is also of a certain interest in this kind of study.

    The information collected could be included in the discharge report and benefit the patient's general practitioner, with information on the oral situation and any treatment required, (e.g. through replacement therapy and denture repairs) to facilitate and support the general rehabilitation of the patient is holding. 

The manuscript reads easily and clearly. A recent reference from Doshi in 2022 to be included in the bibliography would update the article. Another reference from 2021 on the night-time wearing of this type of denture in an institution would also be desirable. The result presented in the article corresponds to treatment in the Japanese context. It seems that we do not find such results in England in particular. The data presented in the article are not sufficiently detailed concerning the patients (sex, general local diseases, hygiene, class of edentulous, type of denture lost, maxillary, mandibular, removable stabilized on implants, adhesive).

 Comments:

 Abstract.

ligne 13 : Materiel and methods. Can you cantify this improvement? specifying that you used a questionnaire only.

Ligne 22: can you cantify this improvement?

Introduction.

    Ligne 42: however the results presented by Harrison are clearly superior to those of the article.

 Ligne 89: More precision on these two patients is desirable in order to enrich the article.

 Ligne 124: a recent article by Doshi 2022 referenced at this level is desirable.

Ligne 152: sleep apnea in denture wearers is also to be taken into account in the elderly. (recent article on the subject)

Effects of nocturnal wearing of dentures on the quality of sleep and oral-health-related quality in edentate elders with untreated sleep apnea: a randomized cross-over trial

Elham Emami, et al . 2021 Oct; 44(10).

Ligne 207:the result obtained in this article is undoubtedly a reflection of the good management of patients in institutions in Japan. However, this single-center study requires further investigation to confirm these initial results. Unfortunately, there are different levels of benefits for the care of these patients depending on socio-economic conditions. There are also reservations before generalizing this result.

Author Response

Thank you for your helpful review. The points that the reviewer raised have been corrected as follows.

Responses to the comments.

Article: Current denture loss in geriatric felicities.

The objective of this article is to update the data concerning the loss of dentures in retirement homes and specialized establishments welcoming mentally impaired elderly patients. It sheds light on the level of care for these elderly patients living in institutions in Japan.

However, the information concerning the patients included in the study is insufficient.

A questionnaire and the consultation of patient files do not allow to realize exactly the oral state of patients. Also in order to enhance this study, the collection of available medical data would be desirable, supplemented by a mini dental check-up. This as part of overall patient care. Thus, following this examination, the finding of the actual loss of the dental prosthesis remains necessary On the other hand the effective appreciation of the functional value and the aesthetics of the prosthesis of use is also of a certain interest in this kind of study.

    The information collected could be included in the discharge report and benefit the patient's general practitioner, with information on the oral situation and any treatment required, (e.g. through replacement therapy and denture repairs) to facilitate and support the general rehabilitation of the patient is holding. 

The manuscript reads easily and clearly. A recent reference from Doshi in 2022 to be included in the bibliography would update the article. Another reference from 2021 on the night-time wearing of this type of denture in an institution would also be desirable. The result presented in the article corresponds to treatment in the Japanese context. It seems that we do not find such results in England in particular. The data presented in the article are not sufficiently detailed concerning the patients (sex, general local diseases, hygiene, class of edentulous, type of denture lost, maxillary, mandibular, removable stabilized on implants, adhesive).

 Comments:

 Abstract.

ligne 13 : Materiel and methods. Can you cantify this improvement? specifying that you used a questionnaire only.

Ligne 22: can you cantify this improvement?

This study is a pilot study prior to a larger study. Therefore, for the sake of accuracy, this study, although small in size, is limited to subjects at affiliated facilities of our institution. All residents underwent regular oral examinations by us. The results showed that there were two patients with missing dentures. The missing dentures were noted in the logbooks, suggesting that a large retrospective study could be conducted in the future.

Although there are many background factors, we believe that the low incidence of denture loss is largely due to the efforts of the facility caregivers.

Introduction.

Ligne 42: however the results presented by Harrison are clearly superior to those of the article.

The quote from Harrison has been changed to a quote from Doshi et al. (Line 42)

 Ligne 89: More precision on these two patients is desirable in order to enrich the article.

According to the suggestion of the reviewer, I have added details of two residents with denture loss in Results section (Line 98).

 Ligne 124: a recent article by Doshi 2022 referenced at this level is desirable.

The article by Doshi et al. (2022) was cited where appropriate in Introduction section and Discussion section.

Ligne 152: sleep apnea in denture wearers is also to be taken into account in the elderly. (recent article on the subject)

Effects of nocturnal wearing of dentures on the quality of sleep and oral-health-related quality in edentate elders with untreated sleep apnea: a randomized cross-over trial Elham Emami, et al . 2021 Oct; 44(10).

In accordance with reviewer’s suggestion, I have added a sentence related to sleep apnea and dentures (Line 201).

Ligne 207:the result obtained in this article is undoubtedly a reflection of the good management of patients in institutions in Japan. However, this single-center study requires further investigation to confirm these initial results. Unfortunately, there are different levels of benefits for the care of these patients depending on socio-economic conditions. There are also reservations before generalizing this result.

In accordance with reviewer’s suggestion, I have added a statement about the position of this study with respect to countries other than Japan (Line 214).

We have revised this paper to the best of our ability. Thank you for your review and the helpful suggestions. We hope that the revised version of the manuscript will now be considered suitable for publication in medicines.

Sincerely,

Round 2

Reviewer 2 Report

Following the corrections made to the article Title: Current denture loss in geriatric facilities Authors: Miki Endo, Shunsuke Hino *, Miki Yamada, Yosuke Iijima, Nami Nakayama, Kiyoko Ariya, Norio Horie, Takahiro Kaneko. I see no harm in posting it. I thank you for your trust.